# The Usability Testing of VR Interface for Tourism Apps

**Yu-Min Fang \*** and **Chun Lin**

Department of Industrial Design, National United University, Miaoli 36003, Taiwan
\* Correspondence: FanGeo@nuu.edu.tw; Tel.: +886-37-38-1664

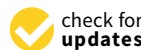

**Featured Application: Human–Machine Interaction/Virtual Reality and Entertainment**

**Abstract:** Virtual reality (VR) is considered to be an emerging technology. This study compared the usability differences of VR travel software, such as Google Street View, VeeR VR, and Sites in VR, for mobile phones. In the pilot study, three post-graduate students and one interface expert were invited to participate in the designed experimental tasks to provide opinions on the first draft of the questionnaire. Next, thirty college students were recruited to join the formal experiment. After operating the VR interface, they were asked to fill out the questionnaire, and a semi-structured interview was conducted. The results are described as follows: (1) Intuitive operation is required to allow people to select objects smoothly; (2) the chosen object requires a feedback mode to inform the user that the object has been selected; (3) the speed of the feedback mode should be adjustable to fulfil the needs of most people; (4) the contrast of icon color needs to be improved to ensure the most efficient verification of the operations; and (5) a search button or reminder function can be added to aid first-time users.

**Keywords:** virtual reality; tourism; usability; user interface

## 1. Introduction

Tourism can enrich our knowledge and broaden our vision. The rise of globalization and the development of transportation facilities have made tourism to become a more popular activity. However, tourism is often affected by many factors, such as time, money, climate, or unanticipated situations in tourist destinations, all of which affect the mood of tourists.

Virtual reality (VR) has been regarded as a recently emerging technology. Its most significant feature is its ability to allow people to enter an immersive presence, and this immersive experience can bring a new application to tourism. Users can use a VR's hardware and software devices to experience and explore local destinations regardless of time and space. Different companies offer software with varied services as well as varying interfaces and applications. Therefore, determining which VR travel software interface for mobile phones will affect the users' inclination to travel to a local area is significant. Is the operation mode of VR travel software interface too complex? For example, does a failure to find the icon arouse the users' disgust? Are users satisfied with the presentation and operation mode of the interfaces of various VR travel application software?

In order to evaluate and compare the interface of VR travel application software, a usability and feasibility study can be utilized. Feasibility can be defined as whether the implementation of these tourism apps was easily and conveniently done; usability can be described as whether these apps could be used to adequately function in a way that enhanced productivity or led to unproductive tasks due to errors. The feasibility of using the apps can be evaluated by considering the time taken for operations, the errors, and subjective feedback from users; the usability can be evaluated by learnability,

intuitiveness, and ease of use [1,2]. In previous studies, other common feasibility evaluation indexes included acceptance, adherence, compliance, motivation to use, user' preferences, system usage, user–system interactions; other usability indexes have included attention, conceptual model, efficiency, interaction, invisibility, impact, memorability, trust [3–6]. Usability and feasibility occasionally share the same evaluation indexes, such as satisfaction, task completion rate, user error rate, and experience (positive or negative) with the system.

Previous studies have reported some significant usability problems with current VR systems [7,8]. Some scholars attempted to study these usability issues in different VR applications. Carol et al. indicated that it is necessary to investigate the usability of these VR technologies as well as the ergonomic constraints. They performed a usability test and observed both users' verbal and nonverbal behaviors [9]. Madathil and Greenstein utilized subjective ratings and questionnaires to test the dependent variables included the time taken to complete the tasks, usability defects, workload and satisfaction degree [7]. Pereira et al. indicated that the VR platform required improvement such as image quality, safety information displayed, and user interface interactions [10]. Hald et al. used the System Usability Scale to test different types of VR setups in terms of user accuracy and speed [11]. Boletsis and McCallum also tested the usability of an augmented reality system using the System Usability Scale, as well as by utilizing user observations and remarks documented by open and semi-structured interviews [12]. In the latest literature, Liu et al. developed an evaluation framework to evaluate the usability of VR interfaces [13], and Amano et al. designed a testbed to test the usability and performance of such mobile apps in in-situ VR environment mobile apps [14].

There has been some research conducted on VR interactions in different venues; however, little work was done on the VR travel software interface. This paper aims to explore the VR travel software interface for mobile phones, compare the differences among various interfaces in use, explore the usability by conducting an experiment (including user interaction satisfaction and ease of use), and compare participants' satisfaction with various VR travel interfaces for mobile phones.

## 2. Literature Review

### 2.1. Virtual Reality and Product Categories

Virtual reality is alternately known as virtual environment. Based on its early stage concept, VR is described as an environment simulated by computer [15–17]. It is a three-dimensional virtual world that can be accessed through the use of computer simulation, which provides users with multiple sensory stimulation. Users feel as though they are actually experiencing the scene and can observe things in three-dimensional space instantly and indefinitely. In addition, users can interact by means of various induction devices (e.g., 3D glasses and 3D gloves) and virtual objects in the scene. When the user is moving, the computer can immediately carry out complex calculations and send back accurate three-dimensional images to create a sense of presence.

A VR display is used to experience virtual reality. Presently, VR displays by different companies have their own characteristics on the market, and these can be categorized into a professional head-mounted display and a smart mobile phone display. The professional head-mounted display types include Sony PlayStation VR, HTC Vive, and Oculus Rift, which are high-specification products designed to be equipped with high-level computers or related equipment and are more expensive. Meanwhile, the smart mobile phone display includes Google Cardboard VR, Samsung Gear VR, and Google Daydream, which can be used through smart mobile phones, particularly mobile phones with specified requirement; these are cheaper than the head-mounted displays.

Using a professional head-mounted display, virtual reality can be experienced through computers complemented by special peripheral equipment and the application software in the related software platform. Based on the classification of virtual reality travel games collected from the Steam software platform, the software can be categorized as either free or paid software. In addition, by searching and sorting travel game application software recommended by the Google Play store, they can be classified

into three categories, namely 360° around-view photos, 360° around-view films, and interactive 360° around-view images.

## 2.2. Usability Testing

Usability is related to the common psychological cognition, behavioral patterns, and cultural connotation of human beings, all of which can be proven through experiments and tests. Usability metrics refer to the data collected to describe and measure interface usability including learnability, ease of use, flexibility, and attitude [18]. These usability measures are described as follows:

Learnability: The time and effort users require to achieve effectiveness of a given task (also called as extent of ease to learn). This includes the time required by a particular group of users to complete a particular job, the number of errors made in completing the job, and the time spent in referencing the system files.

Ease of use: The time experienced users spent in completing certain jobs or the speed of job execution and the mistakes made by users.

Flexibility: The extent that users can adapt to new interaction mode as they become more familiar with the system.

Attitude: The positive or negative feeling of the users toward the system.

The recent literature has shown that usability testing has been utilized for evaluating the VR interface. Questionnaires for User Interaction Satisfaction (QUIS) and the System Usability Scale (SUS) are the two commonly used usability test scales. Altarteer et al. and AlFalah et al. adapted QUIS to test their proposed VR interface [19,20]; Whitney et al. and Gu et al. provided SUS scores to prove the effectiveness of their proposed VR systems [21,22].

## 2.3. Usability Test Scale

The two usability test scales are described as follows:

QUIS: Proposed by the Human–Computer Interaction Lab, University of Maryland, USA, QUIS is used mainly to measure the subjective satisfaction of system users in regard to a human–computer interface. The information, visibility, learning ability, and system functionality of the measurement system, as well as the questionnaires and scale, can be modified according to the needs of the research institute [23–25]. All questions are evaluated by a seven-item Likert scale on a scored range of 1–7, where 1 represents very dissatisfied; 4 represents neutral; and 7 represents very satisfied. QUIS was modified to a five-point Likert scale in this study for testing VR interfaces; "Overall response," "Interface representation," "Interface information," and "learnability" were the only categories used.

SUS: Proposed by John Brooke in 1986, this is the most commonly used questionnaire scale to evaluate ease of use of a system and has also been widely used in fast tests of system interfaces, desktop programs, and website interfaces [26,27]. SUS is recognized as a fast scoring test tool for ease of use. Experts' study results on SUS suggest it can also be used in the quantitative analysis of a small sample. This tool is also conducted using a five-point Likert scale, and the perceptions are scored on a scale of 1–100.

## 3. Materials and Methods

### 3.1. Research Design and Subjects

A preliminary questionnaire and experimental tasks were designed, and a pre-test was conducted subsequently. Under the experts' guidance, the questionnaire and experimental contents were revised according to the feedback from the pre-test. In the formal experiment, 30 subjects were gathered and asked to take part in the final three-interface experiment and fill out questionnaires. During the experiment, the subjects were asked to put on a head-mounted display (VR Box 2017, CeoMate Technology Co., Ltd., Taiwan) with a smart mobile phone (Sony Xperia X 2016, Sony Corporation, Japan) to operate the VR travel application and complete the tasks (Figure 1). ApowerMirror (ApowerMirror

2018, Apowersoft Ltd., Hong Kong, China), a screen mirroring application, was used to stream subjects' screen on the smartphone to Windows computer (Acer Notebook computer, VR Ready). The smartphone was also remotely intervened and controlled on the notebook computer using mouse and keyboard if necessary. Finally, the subjects were invited to fill out the questionnaire, answer the usability scales (QUIS and SUS), provide personal data, and complete their interviews.

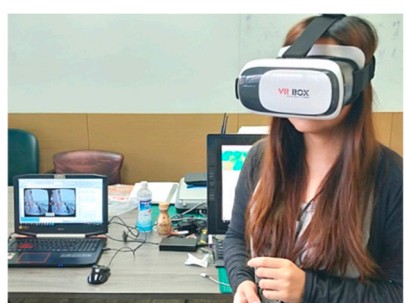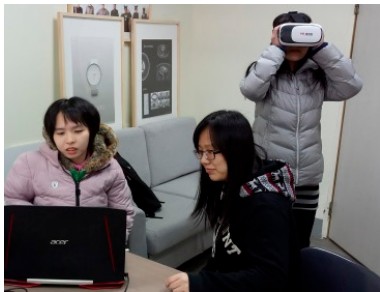

**Figure 1.** The subjects were asked to put on a virtual reality (VR) Box with a smart mobile phone to operate the VR travel application and complete the tasks.

*3.2. Materials*

The subjects were asked to put on the VR devices to operate three kinds of travel apps, including Google Street View, VeeR VR, and Sites in VR. Table 1 lists the different attributes of these apps, including type, usage mode, operating mode, operational diversity (VR mode), ease and difficulty of use, information volume (content), diction (VR interface), sound, and tourist destination for the experiment.

**Table 1.** Sample table for formal experiment—VR travel app for mobile phones.

| Name | Google Street View | VeeR VR | Sites in VR |
|---|---|---|---|
| Pictures | | | |
| Type | 360° round-view photos | 360° round-view film | 360° round-view images |
| Usage mode | 1. Open the app. 2. Select an option. 3. Put the mobile phone on a display. 4. Experience the 360° round-view. 5. Control the front and back directions through a Bluetooth remote controller. 6. Take the mobile phone out. | 1. Open the app. 2. Select option. 3. Put the mobile phone on a display. 4. Experience the 360° round-view. 5. Focus on the selected film. 6. Take the mobile phone out. | 1. Open the app. 2. Put the mobile phone on a display. 3. Focus on the selected option. 4. Experience the 360° round-view. 5. Focus on other places. 6. Take the mobile phone out. |
| Operating mode | Bluetooth game rocker VR BOX CASE mobile controller | Fixation point (watch for a while) | Fixation point (watch for a while) |
| Operational diversity (VR mode) | Low (no choice) | Middle (able to select the length of the film) | High (able to select other positions) |

**Table 1.** *Cont.*

| Name | Google Street View | VeeR VR | Sites in VR |
|---|---|---|---|
| Ease of use | Simple | Difficult | Moderate |
| Information volume (content) | More (global) | More (diversified) | Less (only particular places) |
| Diction (VR interface) | Less (No textual description) | Less (No textual description) | Less (No textual description) |
| Sound | No | Yes | No |
| Tourist destination | Eiffel Tower | Eiffel Tower | Eiffel Tower |

All subjects were asked to operate three apps. In order to avoid the practice effect in VR operation, thirty subjects were randomly divided into six groups. Each group was assigned to perform one of the six combination order of operating three apps.

### 3.3. Questionnaire Design

A quantitative design was adopted to develop a three-part questionnaire (Table 2). The first part focused on respondents' demographics and product experience. The second part was the main questionnaire. (1) The QUIS (containing 13 items after revision) was revised to a five-point scale to measure the respondents' interface usability satisfaction. (2) The five-point SUS was used to measure perceived usability. Respondents' scores were converted to a scale of 1–100.

**Table 2.** Questionnaire content.

| Items | Number of Items | Items | Content |
|---|---|---|---|
| 1. Basic information and background | 12 | Multiple choice | Respondents' backgrounds |
| 2. QUIS | 13 | Five-point SD scale | Respondents' satisfaction: 2.1 Overall response 2.2 Interface representation 2.3 Interface information 2.4 Interface learnability |
| 3. SUS | 10 | Five-point Likert scale/converted to 1–100 scale | Perceived interface usability |

In the pre-test, three interface experts suggested that original seven-point scale of QUIS was too subtle for subjects to distinguish the differences. Some questions were also not relevant to the VR interface anymore. Therefore, QUIS was modified to a five-point Likert scale without inappropriate questions.

In addition to the formal survey, a semistructured interview was conducted to explore the explanations for the survey results. The interview consisted of three parts: (1) The respondents' consideration in preferences towards these three types of interfaces; (2) the respondents' suggestions to revise the interface and their reasons; and (3) the respondents' opinions to the usability.

## 4. Results and Discussion

### 4.1. Background Analysis of Subjects

In this section, the subjects' personal background, their tourism experience, and their experience in using devices and VR devices are explored. A total of 30 students from National United University are gathered as subjects. Table 3 shows the subjects' gender and educational background. Questions

on whether the subjects had accessed tourism websites before were explored. The answer choices included "have not accessed," "rarely accessed," "commonly accessed," "often accessed," and "daily access," from four, three, four, thirteen, and six subjects, respectively. The population proportions were 13.3%, 10%, 13.3%, 43.3%, and 20%, respectively.

**Table 3.** Questionnaire population structure table.

| Items | Gender | Amount | Percentage (%) |
|---|---|---|---|
| Gender | Male | 15 | 50% |
| | Female | 15 | 50% |
| Education | University | 25 | 83.3% |
| | Institution | 5 | 16.7% |

Based on the data above, 76.6% of the subjects had experienced accessing tourism websites at the level above "common" (including "common," "often," and "daily access"), whereas the subjects who "have not accessed" and "rarely accessed" tourism websites accounted for 23.4% of the subjects. The subjects who enjoyed tourist destinations through Google map and pictures or films on the internet at above the "common" level accounted for 73.3% of the respondents, whereas 26.7% reported they "have not accessed" and "rarely" accessed tourism websites. These results show most people will search for tourist destinations by using tourism-related equipment.

Meanwhile, 56.6% of the subjects adopted new technology VR Products at above the "common" level, whereas 43.4% of the subjects "have not accessed" and "rarely" access tourism websites. Subjects who used or did not use VR products were about even. Moreover, subjects who enjoyed tourist destinations using VR travel apps at above the "common" level accounted for only 20%, whereas subjects who "have not accessed" and "rarely access" accounted for 80%. These results suggest that VR products have not been used widely by college students. Therefore, VR products are not often used when searching for tourism-related information.

### 4.2. Usability Analysis: QUIS

A QUIS was used in this paper to realize the subjects' satisfaction with the interface after using the three kinds of interfaces. The scale is measured by five-point Likert scale, where the score closest to 5 means that the subject is more satisfied and the score closest to 1 means the subject is more dissatisfied. QUIS was adopted to understand the differences among the three interfaces based on overall response, interface representation, interface information, and learnability. ANOVA statistical results are shown in Table 4 and Figure 2.

**Table 4.** Mean and standard deviation of Questionnaires for User Interaction Satisfaction (QUIS). (Standard deviation listed in brackets. Likert scale 1–5).

| Group | Google Street View | VeeR VR | Sites in VR |
|---|---|---|---|
| Overall response | 4.03 (0.72) | 3.37 (1.00) | 4.03 (0.89) |
| Interface representation | 3.97 (0.72) | 3.33 (1.06) | 3.97 (1.03) |
| Interface information | 3.60 (0.72) | 3.07 (1.01) | 3.83 (0.87) |
| Interface learnability | 4.30 (0.88) | 3.80 (0.96) | 4.13 (0.86) |

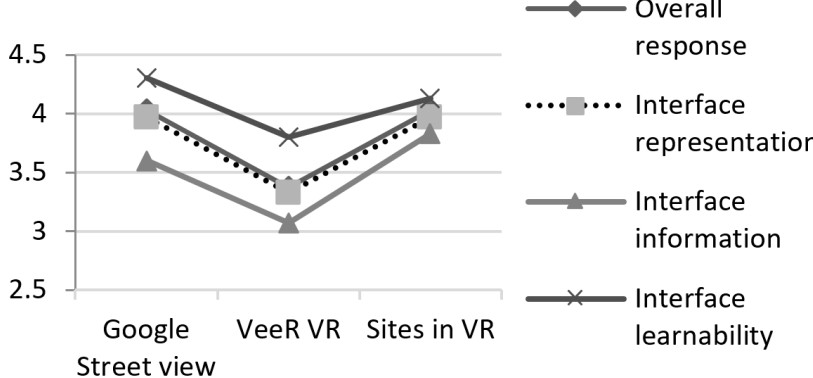

**Figure 2.** QUIS comparison of the three interfaces (unit: Points; five-point Likert scale).

The statistical results of these four usability measures are described as follows:

*Overall Response:* The mean result of this item in the interfaces of Google Street View and Sites in VR tended to be positive (M > 4), whereas that of VeeR VR tended to be negative (M < 4), as shown in Figure 2. The results of the ANOVA showed a significance relationship between VeeR VR and the other two interfaces, whereas no significance was observed between Google Street View and Sites in VR. No difference was also observed between these two interfaces in terms of the overall response, and both were significantly higher than VeeR VR, as illustrated in Table 5.

**Table 5.** Post-hoc test of QUIS: The significance among the three interfaces.

| | | | |
|---|---|---|---|
| **Overall response** | | | |
| **Significance** | **Google Street View** | **VeeR VR** | **Sites in VR** |
| Google Street View | | 0.016 * | 1.000 |
| VeeR VR | 0.016 * | | 0.016 * |
| Sites in VR | 1.000 | 0.016 * | |
| **Interface representation** | | | |
| Significance | Google Street View | VeeR VR | Sites in VR |
| Google Street View | | 0.040 * | 1.000 |
| VeeR VR | 0.040 * | | 0.040 * |
| Sites in VR | 1.000 | 0.040 * | |
| **Interface information** | | | |
| Significance | Google Street View | VeeR VR | Sites in VR |
| Google Street View | | 0.069 | 0.591 |
| VeeR VR | 0.069 | | 0.005 * |
| Sites in VR | 0.591 | 0.005 * | |
| **Interface learnability** | | | |
| Significance | Google Street View | VeeR VR | Sites in VR |
| Google Street View | | 0.105 | 0.774 |
| VeeR VR | 0.105 | | 0.362 |
| Sites in VR | 0.774 | 0.362 | |

* represents statistical difference; $p < 0.05$.

*Interface Representation:* The mean result of this item in the interfaces of Google Street View, VeeR VR, and Sites in VR tended to be negative (M < 4), as shown as Figure 2. The results of the ANOVA showed a significant relationship between VeeR VR and the other two interfaces, whereas no significant relationship was found between Google Street View and Sites in VR. It also showed that there was no difference between the two interfaces in terms of interface representation, and both interfaces were significantly higher than VeeR V.

*Interface Information:* The mean result of this item in the interfaces of Google Street View, VeeR VR and Sites in VR tended to be negative (M < 4), as shown in Figure 2. ANOVA analysis result showed the existence of a significant relationship between VeeR VR and Sites in VR, whereas no significant relationship was found between Google Street View and other two. It also showed that no significant relationship existed between Google Street View and other two interfaces in the interface information.

*Interface Learnability:* This item's mean result in the interfaces of Google Street View and Sites in VR tended to positive (M > 4), whereas VeeR VR tended to be negative (M < 4), as shown in Figure 2. The results of the ANOVA showed that no significant difference could be observed between the three interfaces, while all of them were significantly higher than VeeR VR (Table 5).

The data above show that Google Street View and Sites in VR significantly scored the highest in terms of overall response and interface representation. Google Street View reached the highest score in terms of interface learnability, while Sites in VR had the highest in interface information. Google Street View and Sites in VR exhibited high interface overall response and interface learnability, whereas VeeR VR tended to be negative in all items. It is speculated that finding and selecting menu icons to navigate through a tour scene is easy because the interface function of the Google Street View is relatively simple. Sites in VR retains continuity of the interface design, making it easy for users to find the correct operations even if their first choice selected is wrong. The information on VeeR VR screen is displayed too fast, making it challenging to find the interface icon and causing confusion, thereby creating an impression that it is difficult to operate.

### 4.3. Usability Analysis: SUS

To realize ease of use of the three interfaces, the questionnaire was used for a survey and was measured by the SUS scale, which is a five-score scale. SUS consists of negatively and positively worded items. For statistical calculation, the negative items were reversed. A selection closer to 5 means that it is more positive, whereas a selection closer to 1 is more negative.

In this paper, a SUS scale was used to test and therefore understand the indexed ease of use of the three interfaces. The ANOVA statistical analysis results are shown in Table 6 and Figure 3.

**Table 6.** Mean value and standard deviation of System Usability Scale (SUS) in three forms of interface (Standard deviation listed in brackets, the full score is 100).

| Group | Google Street View | VeeR VR | Sites in VR |
|---|---|---|---|
| SUS | 76.92 (18.77) | 54.83 (22.86) | 76.58 (19.53) |

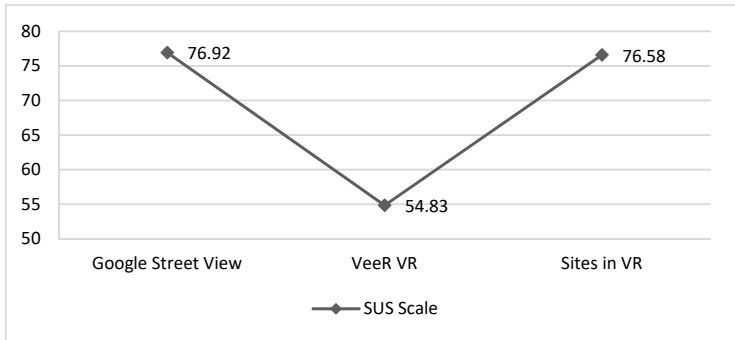

**Figure 3.** Comparison among the three interfaces in SUS scale (full score is 100).

Google Street View and Sites in VR scored both higher than 70 in the interface, which implies that they had a better usability than VeeR VR, as shown in Figure 3. Meanwhile, the VeeR VR interface had a lower mean value (around 50), and its interface design needs to be improved. The results of ANOVA show the VeeR VR differed significantly from Google Street View and Sites in VR, and no significant difference could be observed between Google Street View and Sites in VR. This finding indicates that both Google Street View and Sites in VR have similar usability, and both are better than VeeR VR in terms of interface usability. The results are illustrated in Table 7.

**Table 7.** Post-hoc test of ease of use: Significant differences among the three interfaces.

| Significance | Google Street View | VeeR VR | Sites in VR |
|---|---|---|---|
| Google Street View | | 0.000 * | 0.998 |
| VeeR VR | 0.000 * | | 0.000 * |
| Sites in VR | 0.998 | 0.000 * | |

* Represents statistical difference; $p < 0.05$.

The data above indicate that Google Street View and Sites in VR are better than VeeR VR in SUS scores. This shows that subjects considered Google Street View and Sites in VR to function better than VeeR VR with respect to usability. In addition, the interface of VeeR VR is not straightforward and causes confusion for first-time users to operate. By contrast, Google Street View is considered more user-friendly and easier to use. Its arrow icons serve well for users to easily understand that the arrow icon represents the direction of moving.

As for Sites in VR, while it and VeeR VR share the same operational mode, the former is easier to use. The interface icons of VeeR VR are not straightforward enough; moreover, its selection speed is regarded as too fast, which thus makes it difficult to find the menu.

## 5. Conclusions

This paper aimed to explore the different interface representations and operation modes of VR travel software for mobile phones. An interface design guideline has been successfully proposed for future researchers or designers to utilize in VR interface design. The results are categorized by usability metric and reported as follows: (1) Learnability: Intuitive operation is necessary for first-time users to learn and allow them to select objects or menus smoothly; 3D Icons need to be carefully designed to comprehend and learn quickly. (2) Ease of use: Simple icons for navigating through tour scene need to be carefully designed, the icon color contrast needs to be improved, and the chosen object or menu requires real-time feedback. (3) Flexibility: The speed of the displaying information should be adjustable for users' different response-ability, and a search button or reminder icon can be added to assist first-time users. (4) Attitude: All interface design needs to be examined carefully and fine-tuned to avoid users' confusion or negative emotion.

There are two major limitations in this study that could be addressed in future research. First, to avoid excessive variables, this research only focused on the usability comparison between three VR interfaces without the consideration of sickness effect, hardware capabilities, the formats of VR interfaces, or other potential issues. Second, the results of the interview indicated that the users' emotional response, satisfaction, or technology acceptance degree might be the research directions worthy of further study. Third, the questionnaire items of QUIS and SUS were modified to be suitable for this study. However, these modifications might affect the validity, and this result is only valid for the usability comparison between three VR interfaces in this study.

Finally, future scholars can consider more design variables in the VR system and examine a wide variety of users' response. Designers can also apply the results obtained in this study as a guideline for future designs.

**Author Contributions:** Conceptualization: Y.-M.F. and C.L.; data curation: C.L.; formal analysis: C.L.; funding acquisition: Y.-M.F.; investigation: C.L.; methodology: Y.-M.F. and C.L.; project administration: Y.-M.F.

**Funding:** This research was funded by the Ministry of Science and Technology of Taiwan, grant number MOST 107-2410-H-239 -012 -MY2.

**Conflicts of Interest:** The author(s) declared no potential conflicts of interest with respect to the research, authorship, and/or publication of this article.

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
