# Peer review of "The Usability Testing of VR Interface for Tourism Apps"

_applsci, doi:10.3390/app9163215_

Round 1

Reviewer 1 Report

The research article is focused on the testing of VR Interface for tourism apps. The article compares several available applications used by the VR headset with mobile phones supporting.

This paper is aimed at comparing several available applications (Google street, Sites and VeeR VR) and comparing them with the evaluation using Usability analysis: QUIS. In my opinion, very basic research is done in the article. Several respondents were involved in the research, but I would expect more extensive processing of the conclusion and future research section.

Author Response

Dear Reviewer,

Thanks for the valuable comments and offering the opportunity for us to improve this paper. Please check the attachment for our response.

If you have any further comments, please feel free to tell me. I want to do my best to make this paper better. Your help is much appreciated.

Reviewer 2 Report

This article presents an interesting research project, in my opinion, if it could be improved. From my point of view, this paper could definitely bring its originality to its research field through an interesting theme in aspects of virtual reality and/or mobile tourisms. However, this article is insufficient to be publishable yet in its present format mainly because of the following reasons:

- The introduction part lacks situating this research in the global context/background. In my opinion, it would better add some literature reviews stating similar previous attempts to cover key issues including core development procedures of virtual/mixed reality and major evaluation processes for its performance.

- In the main chapter, then, this paper needs to clarify usability, feasibility and/or applicability with main evaluation indexes by supporting clues from the references. And, it could be commented if the methodology proposed in this study could be useful to compare additional similar applications in points of VR interfaces for tourism from other venues.

- English needs to be verified in overall (for example, Line 26: ‘has made,’ Line 53: ‘types includes,’ Line 229: ‘is find,’ and so on) and more references are recommended to be reviewed.

Author Response

(The authors gave the same response as above.)

Reviewer 3 Report

applsci-558659

Title: The Usability Testing of VR Interface for Tourism Apps

The authors present the results of a usability user evaluation study of VR interfaces for tourism applications. In the study mobile phones were used to assess the usability of Google Street View, VeeR VR, and Sites in VR applications in VR mode. Recommendations are given.

All the references are accounted for in the manuscript.

To further improve this paper, the authors can consider the following suggestions:

1.     How was the study counterbalanced – i.e. how was the order of different scenarios (i.e. applications) determined?

2.     Which specific devices (i.e. mobile phones) were used?

3.     How was the QUIS questionnaire modified and why (lines 80-89)?

4.     The description in chapter 4.3 SUS (lines 202-205) is incorrect. SUS consists of alternating questions. Some are positive and some negative based. The score 5 does not always indicate the most positive option.

5.     SUS scores are incorrectly interpreted (lines 215-218). SUS score over 70 does not indicate “excellent usability”. See:  Bangor, Aaron, Philip Kortum, and James Miller. "Determining what individual SUS scores mean: Adding an adjective rating scale." Journal of usability studies 4.3 (2009): 114-123.

6.     Were the effects of VR sickness considered or detected during the study?

7.     The authors should improve grammar and expression, e.g. lines 226-223.

Author Response

(The authors gave the same response as above.)

Round 2

Reviewer 2 Report

This paper has a lot improved from its previous version. In my opinion, however, this paper still needs to clarify usability, feasibility and/or applicability by suggesting main common evaluation indexes for a comparative study among similar tools for tourism from other venues.

Reviewer 3 Report

The authors have addressed some of the issues raised, however there still remain:

1.     The expression and grammar should be improved. E.g. line 244-250, expression “It suggests that subjects are with higher confidence in the interface of Google street view and Sites in VR.” is unclear and grammatically incorrect.

2.     Any modifications of the standard methods (e.g. SUS, QUIS, …) should be done with care and if applied, sufficiently explained. The explanation under lines 220-223, “SUS consists of alternating questions. For statistical calculation, the 222 negative based questions had been reversed.” is not scientifically sound. The SUS is a very simple standard usability measurement method and should be used in the original form, as intended. Modifications can possibly affect the validity of the method and this make the comparison of the obtained results with other studies difficult. The authors should clarify this.
